

# Low resolution scans can provide a sufficiently accurate, cost- and time-effective alternative to high resolution scans for 3D shape analyses

Ariel E. Marcy[1], Carmelo Fruciano[2], Matthew J. Phillips[3], Karine Mardon[4,5] and Vera Weisbecker[1]

[1] School of Biological Sciences, University of Queensland, Brisbane, Queensland, Australia
[2] Institut de biologie de l'Ecole normale supérieure, Ecole normale supérieure, Université Paris, Paris, France
[3] School of Earth, Environmental and Biological Sciences, Queensland University of Technology, Brisbane, Queensland, Australia
[4] Centre for Advanced Imaging, University of Queensland, Brisbane, Queensland, Australia
[5] National Imaging Facility, University of Queensland, Brisbane, Queensland, Australia

Corresponding author
Ariel E. Marcy, a.marcy@uq.edu.au

## ABSTRACT

**Background**. Advances in 3D shape capture technology have made powerful shape analyses, such as geometric morphometrics, more feasible. While the highly accurate micro-computed tomography (μCT) scanners have been the "gold standard," recent improvements in 3D surface scanners may make this technology a faster, portable, and cost-effective alternative. Several studies have already compared the two devices but all use relatively large specimens such as human crania. Here we perform shape analyses on Australia's smallest rodent to test whether a 3D scanner produces similar results to a μCT scanner.

**Methods**. We captured 19 delicate mouse (*Pseudomys delicatulus*) crania with a μCT scanner and a 3D scanner for geometric morphometrics. We ran multiple Procrustes ANOVAs to test how variation due to scan device compared to other sources such as biologically relevant variation and operator error. We quantified operator error as levels of variation and repeatability. Further, we tested if the two devices performed differently at classifying individuals based on sexual dimorphism. Finally, we inspected scatterplots of principal component analysis (PCA) scores for non-random patterns.

**Results**. In all Procrustes ANOVAs, regardless of factors included, differences between individuals contributed the most to total variation. The PCA plots reflect this in how the individuals are dispersed. Including only the symmetric component of shape increased the biological signal relative to variation due to device and due to error. 3D scans showed a higher level of operator error as evidenced by a greater spread of their replicates on the PCA, a higher level of multivariate variation, and a lower repeatability score. However, the 3D scan and μCT scan datasets performed identically in classifying individuals based on intra-specific patterns of sexual dimorphism.

**Discussion**. Compared to μCT scans, we find that even low resolution 3D scans of very small specimens are sufficiently accurate to classify intra-specific differences. We also make three recommendations for best use of low resolution data. First, we recommend that extreme caution should be taken when analyzing the asymmetric component of shape variation. Second, using 3D scans generates more random error due to increased

landmarking difficulty, therefore users should be conservative in landmark choice and avoid multiple operators. Third, using 3D scans introduces a source of systematic error relative to μCT scans, therefore we recommend not combining them when possible, especially in studies expecting little biological variation. Our findings support increased use of low resolution 3D scans for most morphological studies; they are likely also applicable to low resolution scans of large specimens made in a medical CT scanner. As most vertebrates are relatively small, we anticipate our results will bolster more researchers in designing affordable large scale studies on small specimens with 3D surface scanners.

## INTRODUCTION

An organism's shape reveals many facets of its biology, including its evolution, ecology, and functional morphology. In the past three decades, geometric morphometrics has revolutionized the field of shape research with better analysis and visualization of shape complexity (*Rohlf & Marcus, 1993*; *Zelditch, Swiderski & Sheets, 2012*). As imaging technology continues to advance, three-dimensional (3D) data have become extremely common in geometric morphometric studies, especially in the cases in which 2D data poorly represent the actual 3D object (*Buser, Sidlauskas & Summers, 2017*; *Cardini, 2014*; *Fruciano, 2016*; *Reig, 1996*). 3D capture methods include very high resolution yet high cost micro-computed tomography (μCT) scanners, which usually require time-intensive sectioning with specialized software. In contrast, 3D surface scanners offer lower acquisition costs as well as faster scanning and processing, but has the disadvantage of generally lower resolution, which limits its use on very small specimens (Fig. 1). For confident use of surface scans in small specimens, it is therefore important to assess the measurement error introduced by choosing a 3D surface scanner for geometric morphometrics.

Most vertebrates would be considered small, for example about two thirds of mammals are below 10 kg (*Weisbecker & Goswami, 2010*), which would translate to small skeletal specimens. Therefore, morphometric studies proposing large sample sizes must be very well funded to use a μCT scanner or have a low-cost option, such as a 3D surface scanner. Previous studies have compared μCT scans to 3D surface scans, however, these were all done in large animals, primarily primates (*Badawi-Fayad & Cabanis, 2007*; *Fourie et al., 2011*; *Katz & Friess, 2014*; *Robinson & Terhune, 2017*; *Sholts et al., 2010*; *Slizewski, Friess & Semal, 2010*). While these studies found low error and high repeatability in 3D surface scans similar to μCT scans, there was a suggestion that higher error occurred in the sample's smaller specimens (*Badawi-Fayad & Cabanis, 2007*; *Fourie et al., 2011*). Other recent studies have conducted 3D geometric morphometric studies on small vertebrate skulls but nearly all have relied exclusively on μCT scanning (*Cornette et al., 2013*; *Evin, Horacek & Hulva, 2011*). The only exception we are aware of is *Muñoz Muñoz, Quinto-Sánchez & González-José (2016)*, which successfully used photogrammetry—a technique

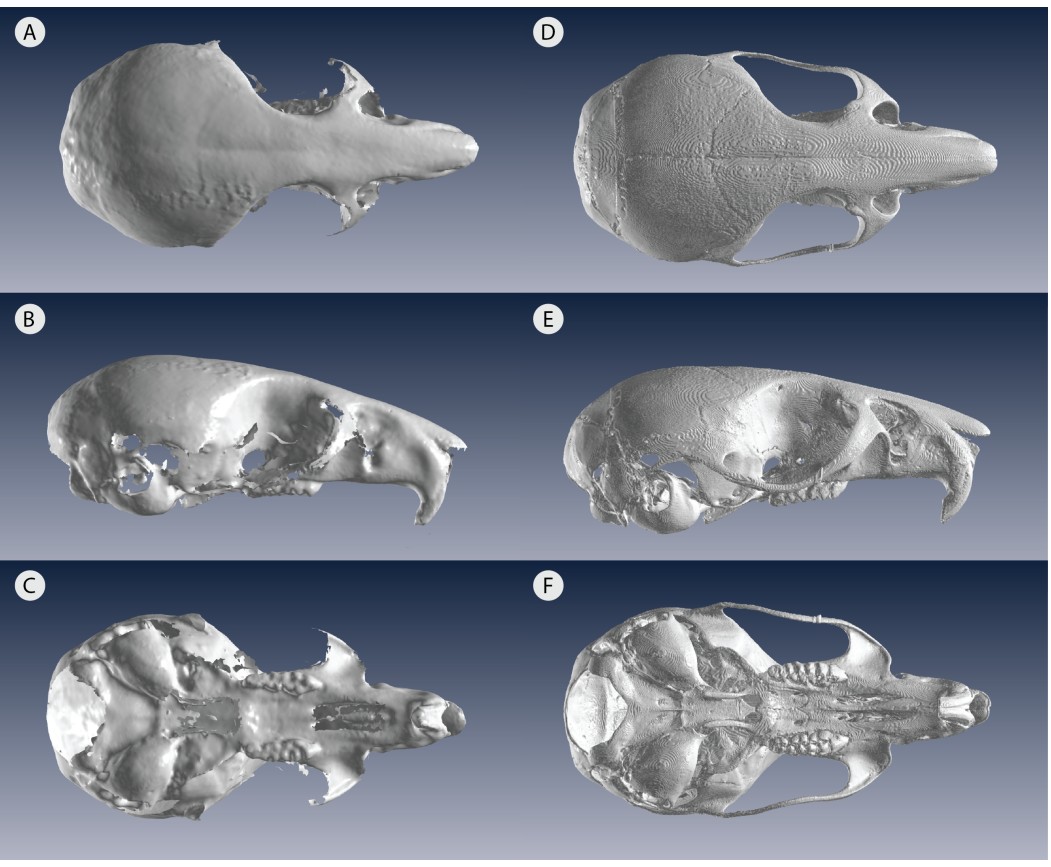

**Figure 1** **Low resolution 3D surface scans compared to μCT scans of the same delicate mouse crania.** 3D scans of (A) dorsal view (B) lateral view and (C) ventral view compared to μCT scans of (D) dorsal view (E) lateral view and (F) ventral view. All crania are rendered in Viewbox v. 4.0.

combining 2D photographs into a 3D model—to analyze domestic mouse skulls, *Mus musculus domesticus* (Linnaeus, 1758). Photogrammetry, like 3D surface scanning, is a low-cost alternative to μCT and comes with its own trade-offs in time and scan resolution (*Katz & Friess, 2014*). Compared to the new generation of blue light surface scanners, photogrammetry requires more time for image acquisition and for file processing (*Katz & Friess, 2014*). A previous study on a single macaque specimen reported inconsistent levels of error across operators and scanners, which contributed to the lack of general pattern for differences across scanners/resolutions (*Shearer et al., 2017*). However, using an interspecific dataset, (*Fruciano et al., 2017*) reported higher repeatability for the higher resolution scans and 2.07–11.26% of total variance due to scan type (depending on device, operator and landmark set combination). We expect that small specimens would exacerbate any variation due to device and the interaction of device with other factors, such as landmark choice and operator. More work comparing these different methods—μCT scanning, 3D surface scanning, and photogrammetry—will allow researchers to make an informed decision. For example, for those with time constraints in museum collections, a fast 3D surface scanner may be the best option if the resolution is suitable for specimen size.

The lower resolution of 3D surface scanners may increase both random and systematic measurement error, which is exacerbated by small specimens because operators may have more difficulty identifying landmark locations (*Arnqvist & Martensson, 1998*; *Fruciano, 2016*). Random error increases variance without changing the mean; this "noise" dilutes biologically informative patterns and, in principle, decreases statistical power (*Arnqvist & Martensson, 1998*; *Fruciano, 2016*). By contrast, systematic error is non-randomly distributed, thus changing the mean and introducing bias to the data (*Arnqvist & Martensson, 1998*; *Fruciano, 2016*). Error assessment can be done with repeated measures of the same individuals (e.g., *Fruciano et al., 2017*; *Muñoz Muñoz & Perpiñán, 2010*; *Robinson & Terhune, 2017*) or by comparison to a "gold standard" or ideal representation of the specimens (*Fruciano, 2016*; *Slizewski, Friess & Semal, 2010*; *Williams & Richtsmeier, 2003*). Repeated measure designs can uncover this systematic error, for example, if one 3D capture method differs from another in a specific, non-random, pattern (*Fruciano, 2016*; *Fruciano et al., 2017*). Furthermore, designs including repeated measures of the same individuals allow partitioning of variance into components, quantifying error due to scan device as compared to biologically-relevant sources of variation such as asymmetry (*Fruciano, 2016*; *Klingenberg, Barluenga & Meyer, 2002*; *Klingenberg & McIntyre, 1998*).

In this study, we quantify the error introduced by studying specimens of a size at the very lower limits of commonly used portable surface scanners' resolution. This situation could also arise when using relatively large specimens, which are nonetheless at the lower limit of a medical CT scanner's resolution for example. We test whether the complex shape of very small specimens can be adequately captured using an HDI109 3D surface scanner (LMI Technologies Inc., Vancouver, Canada) with a stated resolution of 80 μm as compared to a μCT scanner with a resolution of 28 μm. To do so, we use the delicate mouse, *Pseudomys delicatulus* (Gould, 1842), one of the smallest rodents in the world with a 55–75 mm head-and-body length (*Breed & Ford, 2007*). The miniscule *P. delicatulus* crania (∼20 mm) are at the edge of the HDI109 3D surface scanner's range thus providing an extreme test of this scanning device (Figs. 1 and 2). First, we tested whether variation due to scanning device compared to other sources of variation (Fig. 2B). We also asked whether removing asymmetric variation, a common practice in morphological studies when asymmetry is not of interest, changed the results. Second, we tested whether the scanning devices differed in shape variance and in operator error (as measured by repeatability) (Fig. 2C). We also explored how including different types of landmarks impacted repeatability. Finally, we tested whether the shape variation due to scanning device was large enough to impact a small study of intra-specific shape variation using the biologically relevant signal of sexual dimorphism (Fig. 2D).

## METHODS

### Data collection

We selected 19 adult individuals, male and female, of *Pseudomys delicatulus* from the Queensland Museum in Brisbane, Australia (specimen numbers and sexes in Table S1). The cranium from each individual was scanned at the Centre for Advanced Imaging at the

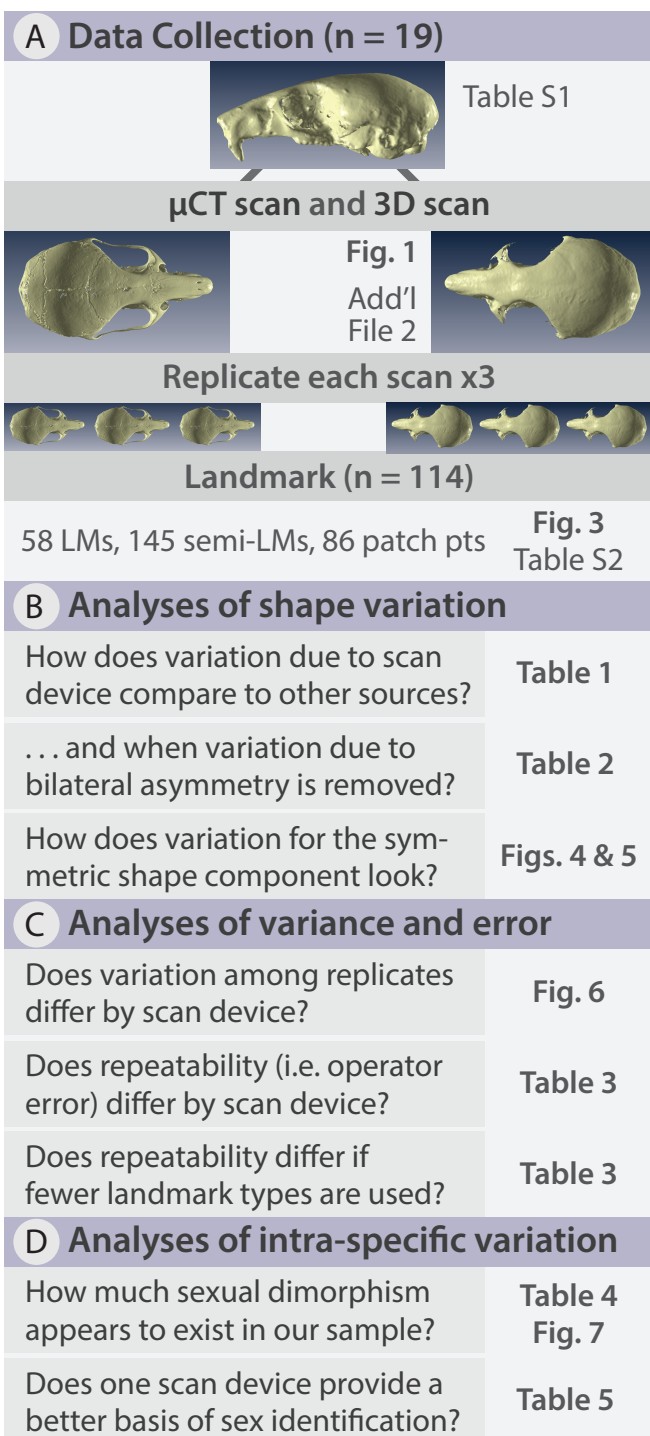

**Figure 2** **Methods flow diagram highlighting the relationship between our questions and our analyses.** (A) All delicate mouse (*Pseudomys delicatulus*) crania were sourced from the Queensland Museum in Brisbane, Australia. Landmarks (LMs) capture homologous points, semi-landmarks (semi-LMs) capture curves between landmarks, and patch points capture surfaces between landmarks and semi-landmarks. (B–D) These sections of questions and associated figure and table numbers summarize how we organize the paper, particularly the Results, into three sets of related analyses.

University of Queensland in a μCT scanner (Siemens Inveon PET/CT scanner, Munich, Germany). The scanner was operated at 80 kV energy, 250 μA intensity with 540 projections per 360°, a medium-high magnification with bin 2 was applied, and 2,000 ms exposure time. The samples were scanned at a nominal isotropic resolution of 28 μm. The data were reconstructed using a Feldkamp conebeam back-projection algorithm provided by an Inveon Acquisition workstation from Siemens (IAW version 2.1, Munich, Germany). Surface models were obtained using Mimics Research version 20.0.

Each cranium was also scanned by a HDI109 blue light surface scanner (LMI Technologies Inc., Vancouver, Canada) with a resolution of 80 μm. For brevity, we will refer to this method as 3D scanning. For this method, the cranium was placed on a rotary table providing the scanner with 360° views. To capture the entire shape, the cranium was scanned in three different orientations: one ventral view with the cranium resting on the frontals and two dorsal views with the cranium tipped to each side, resting on an incisor, auditory bulla, and zygomatic arch. To assist others in replicating our HDI109 3D surface scanning on small specimens, we have included a Standard Operating Procedure with our settings (Supplemental Information 1).

After scanning every individual with both scan methods, we then replicated each 3D model three times so that each individual was represented by six replicates, giving a total sample of 114 3D models (Fig. 2A). Each 3D model was landmarked in Viewbox version 4.0 (dHAL software, Kifissia, Greece; http://www.dhal.com; *Polychronis et al., 2013*). To capture shape, we placed 58 fixed landmarks, 145 semi-landmarks on curves, and 86 patch points (points that during sliding are allowed to slide across a 3D surface defined by the 3D model and semi-landmark borders) for a total of 289 points (Fig. 3, Table S2). We used the template feature in Viewbox to semi-automate the placement of semi-landmarks on curves and to fully automate the placement of patch points. Our landmark design covered most important biological structures except for the zygomatic arch (Fig. 3); we avoided this fine structure because dehydration and loss of support from surrounding muscles during skeletonization almost certainly causes specimen preparation error (*Schmidt et al., 2010*; *Yezerinac, Lougheed & Handford, 1992*).

## Data analysis

The landmark coordinates for all 114 3D models were aligned using a generalized Procrustes analysis followed by projection to the tangent space, as implemented in the R package *geomorph* (v. 3.0.5) (*Adams, Collyer & Sherratt, 2016*; *Adams & Otarola-Castillo, 2013*). Generalized Procrustes analysis of each set of landmark coordinates removes differences in size, position, and orientation, leaving only shape variation (*Rohlf & Slice, 1990*). Semi-landmarks and patches were permitted to slide along their tangent directions to minimize Procrustes distance between 3D models (*Gunz, Mitteroecker & Bookstein, 2005*). The resulting Procrustes tangent coordinates were used as shape variables in all subsequent shape analyses. All our statistical analyses were performed either in R (v. 3.3.3) using the R packages *geomorph* (v. 3.0.5) (Adams 2016; (*Adams & Otarola-Castillo, 2013*) and *Morpho* (v. 2.5.1) (*Schlager, 2017*) or in MorphoJ (v. 1.06d) (*Klingenberg, 2011*).

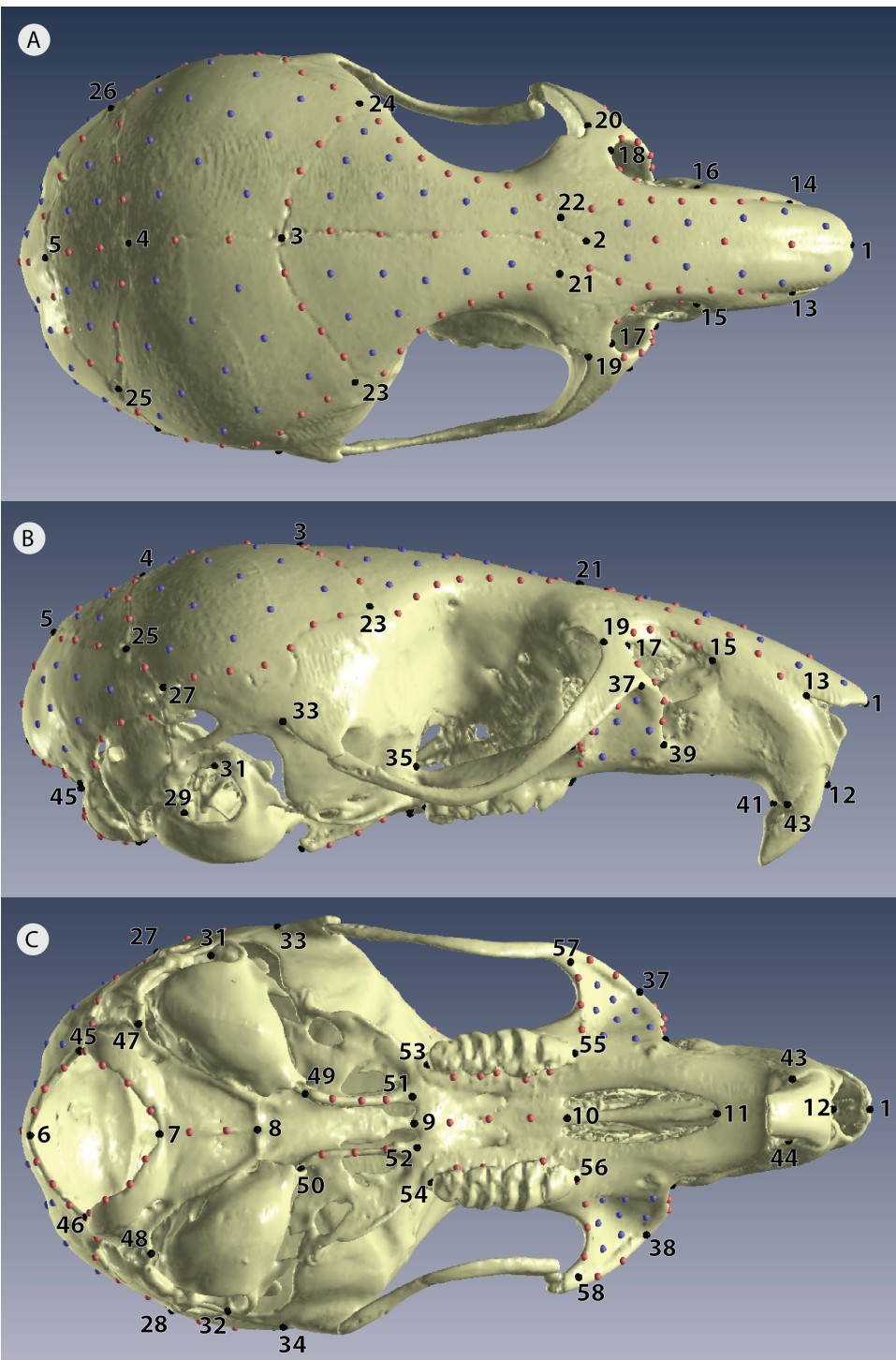

**Figure 3 Positions of landmarks for geometric morphometric analyses.** Locations of fixed landmarks (black points), sliding semi-landmarks (red points) and sliding surface patches (purple points) on a μCT scanned individual. (A) Dorsal view of the cranium. (B) Lateral view. (C) Ventral view. Definitions are given in Table S2.

First, asymmetry is a known source of variation within a sample (*Klingenberg, Barluenga & Meyer, 2002*), so we tested for it with MorphoJ's Procrustes ANOVA function and subsequently removed it (Fig. 2B). Isolating the symmetric component of shape has been undertaken in other 3D surface scanner studies where operator and device error have been of the same magnitude as asymmetric error (*Fruciano et al., 2017*). Variation due to asymmetry is more impacted by operator error because of its smaller effect sizes compared to variation among individuals (*Fruciano, 2016*; *Fruciano et al., 2017*; *Klingenberg et al., 2010*; *Leamy & Klingenberg, 2005*). This suggests that low resolution studies on asymmetry would be negatively impacted. For this reason, we performed most subsequent analyses on the symmetric shape component, with a few exceptions performed for comparison. We then performed a PCA on the symmetric shape variables to visualize the variation between individuals, within scan method replicates, and between scan method replicates. As an exploratory analysis, PCA can help intuitively visualize both random error (greater spread of one scan method replicate compared to the other) and systematic error (repeated pattern of one scan method shifting relative to another). However, further analyses are necessary to quantify these sources of error.

Second, our replicate design allowed us to assess whether an operator digitizing scans from one device was more variable in landmark placement than when digitizing scans from the other device (Fig. 2C). We did so by computing the Procrustes variance for each individual/device combination. In *geomorph*, Procrustes variances are calculated for each set of observations (i.e., replicates) as the sum of the diagonal elements of the set's covariance matrix divided by the number of observations (*Adams, Collyer & Sherratt, 2016*; *Zelditch, Swiderski & Sheets, 2012*). We computed Procrustes variance for each combination of individual and device so that Procrustes variance reflected only variation due to digitization. We then compared Procrustes variance between devices using a box plot and the permutational procedure implemented in *geomorph*. Next we quantified digitization consistency by computing repeatability for each device using the analogue of the intraclass correlation coefficient computed with the Procrustes ANOVA mean squares, as suggested by *Fruciano (2016)*. This value is normally between 0 and 1, with values close to 1 indicating much larger variation due to the factor used in computing the Procrustes ANOVA (in our case, variation among individuals) compared to residual variation (in our case, variation among digitizations). In other words, comparing repeatability between devices gives similar information to that obtained by the box plots of Procrustes variance but on a more easily interpretable scale from 0 to 1. We repeated our computations of repeatability for subsets of the data to test whether introducing semi-landmarks on curves and surfaces (patch points) changed the repeatability relative to a fixed landmark-only dataset. We did so for both 3D and μCT datasets to see if these trends differed by scan device.

Finally, we investigated whether there is a difference between devices in a common task: the correct classification of sexual dimorphism (Fig. 2D). We began with a Procrustes ANOVA in R on the symmetric component for the subset of individuals with sex information ($n = 11$ distinct individuals; $n = 66$ 3D models). This allowed us to gauge the magnitude of the effect of sexual dimorphism compared to other sources of variation, including variation due to scan device. Then with *Morpho*, we averaged the shape of each

**Table 1** **General Procrustes ANOVA on sources of shape variation including asymmetry.** The %Var column of this Procrustes ANOVA demonstrates the relative contribution of each factor to overall variation. It is calculated from the sum of squares for each factor divided by the total sum of squares for all factors.

**(A) All specimens**

|            | *Df*   | SS       | MS       | %Var | *F*  | Pr(>F)  |
|------------|--------|----------|----------|------|------|---------|
| Individual | 8,010  | 6.21E−02 | 7.76E−06 | 48.3 | 11.2 | <.0001  |
| Side       | 415    | 2.37E−02 | 5.70E−05 | 18.4 | 82.4 | <.0001  |
| Ind * Side | 7,470  | 5.17E−03 | 6.93E−07 | 4.02 | 0.54 | 1       |
| Device     | 16,340 | 2.08E−02 | 1.27E−06 | 16.1 | 4.90 | <.0001  |
| Res / Rep  | 65,360 | 1.70E−02 | 2.59E−07 | 13.2 |      |         |

**(B) Only 3D specimens**

|            | *Df*   | SS       | MS       | %Var | *F*  | Pr(>F)  |
|------------|--------|----------|----------|------|------|---------|
| Individual | 8,010  | 3.52E−02 | 4.40E−06 | 51.6 | 4.24 | <.0001  |
| Side       | 415    | 1.31E−02 | 3.15E−05 | 19.2 | 30.4 | <.0001  |
| Ind * Side | 7,470  | 7.75E−03 | 1.04E−06 | 11.4 | 2.79 | <.0001  |
| Res / Rep  | 32,680 | 1.22E−02 | 3.72E−07 | 17.8 |      |         |

**(C) Only CT specimens**

|            | *Df*   | SS       | MS       | %Var | *F*  | Pr(>F)  |
|------------|--------|----------|----------|------|------|---------|
| Individual | 8,010  | 3.45E−02 | 4.31E−06 | 61.7 | 6.41 | <.0001  |
| Side       | 415    | 1.17E−02 | 2.81E−05 | 20.8 | 41.8 | <.0001  |
| Ind * Side | 7,470  | 5.02E−03 | 6.72E−07 | 8.97 | 4.61 | <.0001  |
| Res / Rep  | 32,680 | 4.76E−03 | 1.46E−07 | 8.52 |      |         |

replicate triad for each device, and performed a between-group PCA using sex as group (*Boulesteix, 2005*). Between-group principal component analysis is an ordination technique which is gaining popularity in geometric morphometrics (eg. *Firmat et al., 2012*; *Franchini et al., 2016*; *Franchini et al., 2014*; *Fruciano et al., 2016*; *Fruciano et al., 2014*; *Mitteroecker & Bookstein, 2011*; *Raffini, Fruciano & Meyer, 2018*; *Schmieder et al., 2015*; *Seetah, Cardini & Miracle, 2012*) However, it can be also thought of as a classification tool, as in the *Morpho* implementation which allows performing leave-one-out cross-validation. We, then used cross-validated classification accuracy as a measure of performance in classifying individuals based on their sex.

# RESULTS

## Analyses of shape variation

Our Procrustes ANOVA results indicate that variation among individuals (%Var = 48.3) contributes the most to total variance, with asymmetry (directional and fluctuating), device, and operator error contributing the remainder (Table 1A). The %Var values indicate that directional asymmetry contributes a similar amount of variation as other sources of non-biological variation and that fluctuating asymmetry accounts for much less than digitization error and variation between devices (Table 1A). This means that using

**Table 2  Procrustes ANOVA on the sources of shape variation using the symmetric component of shape.** The R-squared column of this Procrustes ANOVA demonstrates the relative contribution of each factor to overall variation. The shape variation of this dataset is visualized in Figs. 4 and 5.

|  | *Df* | SS | MS | Rsq | *F* | *Z* | Pr(>F) |
|---|---|---|---|---|---|---|---|
| ind | 18 | 6.23E−02 | 3.46E−03 | 0.734 | 25.8 | 21.4 | 0.001 |
| ind:Dev | 19 | 1.24E−02 | 6.52E−04 | 0.146 | 4.86 | 23.7 | 0.001 |
| Residuals | 76 | 1.02E−02 | 1.34E−04 | 0.120 |  |  |  |
| Total | 113 | 8.49E−02 |  |  |  |  |  |

analyses of asymmetry with a combination of μCT and 3D surface scans would likely be unreliable in specimens the size of delicate mice or for specimens scanned at a similarly low resolution. The Procrustes ANOVA results for just the 3D data, confirms this observation in which digitization error is large compared to the components of asymmetric variation (Table 1B). For the 3D dataset, the error term (Res/Rep) contributes 17.8% of variation while asymmetry (Side) contributes 19.2%. In other words, for our 3D scan dataset, error contributes almost as much variation as asymmetry (Table 1B). The Procrustes ANOVA for just the μCT dataset, however, did not have this problem to the same degree. Here, the error term (Res/Rep) contributes only 8.52% of variation while asymmetry (Side) contributes 20.8% (Table 1C). In other words, error contributes less than one half of the contribution of asymmetry in the μCT dataset.

The Procrustes ANOVA on just the symmetric component of shape reports the individual shape variation, representing biological variation, is 73.4% (Table 2). Differences between scan devices represent 14.6% and the residuals encompassing differences among replicates or operator error represent 12.0% of total variance (Table 2). Thus, our Procrustes ANOVA on the symmetric component shows that most of the variation is due to biological sources but the significance of the variation due to device may indicate systematic error.

The PCA on the symmetric component revealed that the first three principal components (PCs) account for 47.0% of total variation (PC1 = 26.4%, PC2 = 12.0%, PC3 = 8.81%, $n = 114$) (Fig. 4). Each of the remaining PCs accounted for 6% or less of total variation therefore we only considered the first three for the exploration of patterns of variation. Positive values along PC1 correspond to a larger braincase relative to the rostrum (Fig. 5A). Positive values along PC2 correspond to a wider frontal bone (Fig. 5B). Finally, positive values along PC3 correspond to a more convex, dorsally-curved ventral surface (Fig. 5C).

The plot of the scores on PC1 and PC2 supports the results from the Procrustes ANOVA on the symmetric component of shape in that most of the visible variation is between individuals, i.e., clusters of each individual's replicates (Fig. 4A). Indeed, regardless of scanning device, replicates from the same individual cluster together (Fig. 4A). For most individuals, replicates occupy non-overlapping regions of the plot except for those around the crowded mean shape near the origin (Fig. 4A). Within each individual's variation on PCA scores, μCT replicates usually form a tighter cluster than the 3D replicates (Fig. 4A). This pattern suggests that using μCT scans introduces less random error than using 3D scans. Furthermore, within an individual, 3D scan replicates tend to cluster closer to other 3D replicates while μCT scan replicates tend to cluster closer to other μCT replicates
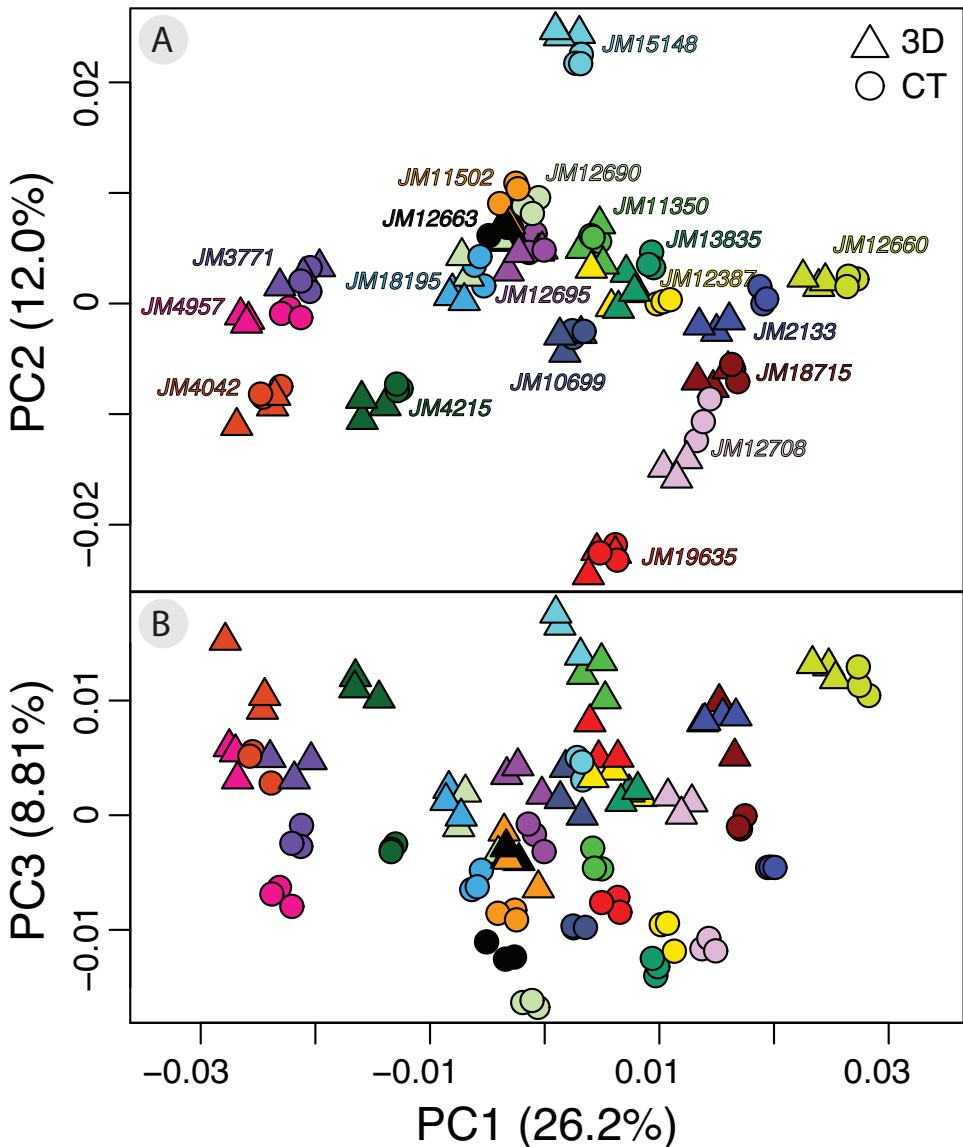

**Figure 4** **Exploratory PCA plots of shape variation showing differences among individuals, scan devices, and replicates of the same scan device.** (A) PC1 versus PC2 and (B) PC1 versus PC3. Each individual has a unique color shared by all of its six replicates. Each individual has three triangles to represent the 3D scanned replicates and 3 circles to represent the μCT scanned replicates. Each axis reports the total variance explained by that principal component.

(Fig. 4A). Indeed, for most individuals, 3D scan replicates score lower than the μCT scan replicates from the same individual on both PC1 and PC2. These results suggest the systematic error may be driven by μCT scans overestimating both braincase volume and frontal bone width relative to 3D scans (Figs. 4A, 5A, 5B).

Overall, the scores along the first two PCs complement and provide an intuitive visualization for the patterns of higher error in 3D scans and of systematic error between

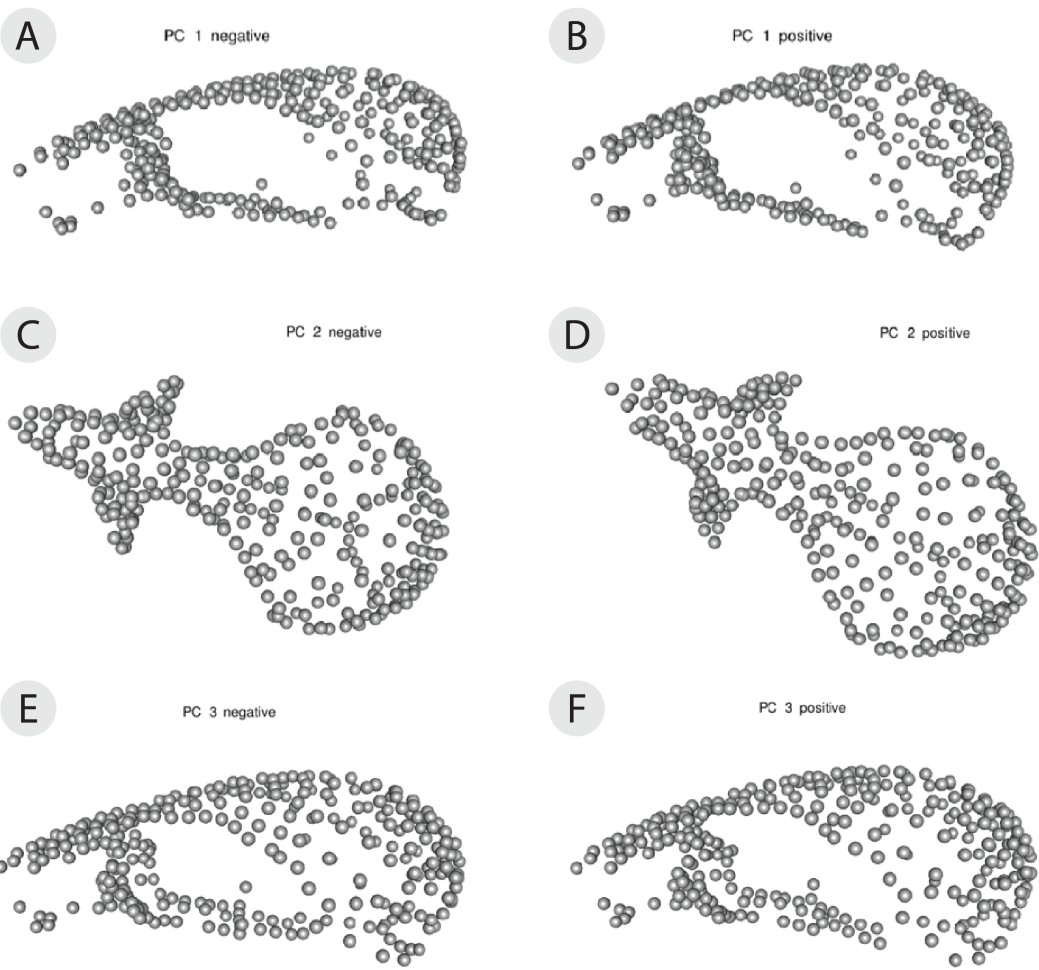

**Figure 5** **3D warp-grids for the three most important principal components, showing minimum and maximum shapes for each PC.** The craniums in (A, C, and E) show the shape of the minimum negative value for each principle component (PC) and the craniums in (B, D, and F) show the shape of the maximum positive value for each PC. Compared to the minimum negative shape (A), more positive values along PC1 (26.4% variance) correspond to a larger braincase relative to the rostrum (B). Compared to the minimum negative shape (C), more positive values along PC2 (11.9% variance) correspond to a wider frontal bone (D). Compared to the minimum negative shape (E), more positive values along PC3 (8.9% variance) correspond to a more dorsally-curved ventral surface (F).

the scan devices as observed in the Procrustes ANOVAs (Tables 1 and 2). The scores along PC1 and PC3 highlight another possible systematic difference between 3D and µCT scans (Fig. 4B). The PC3 axis displaces µCT replicates from 3D replicates such that variation in PC3 scores within individuals is often larger than variation in PC3 scores among individuals (Fig. 4B). On the PC3 axis, almost all 3D scan replicates had higher scores, which correspond to a more dorsally curved ventral surface relative to their corresponding µCT scan replicates (Figs. 4B, 5C).

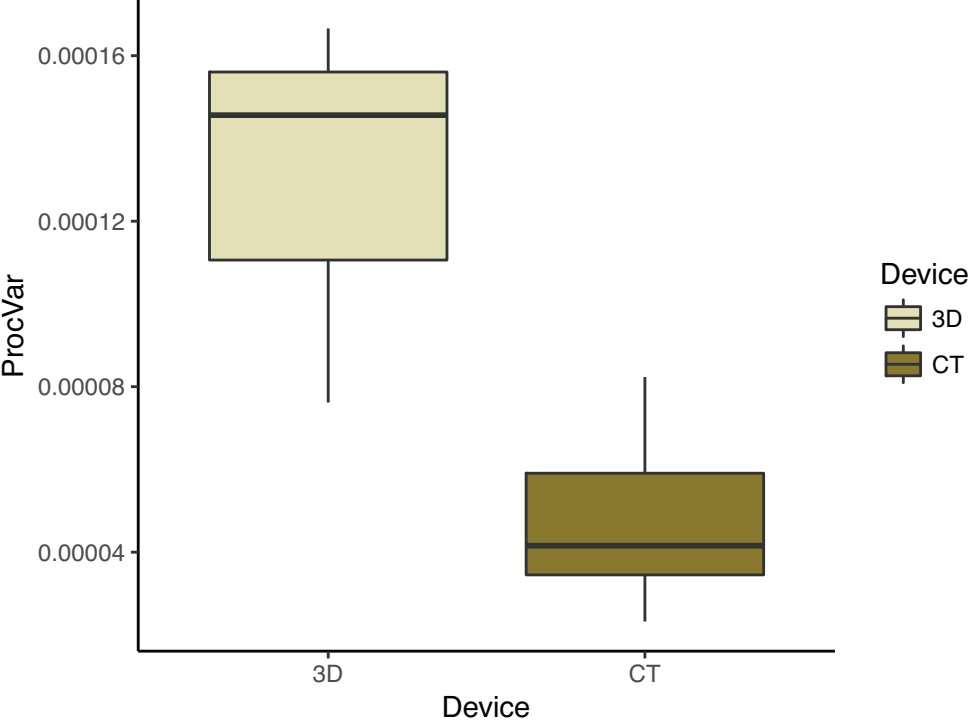

**Figure 6** **Morphological disparity—as measured by shape variation among replicate scan triads—by scanning device reflects operator error.** This box plot summarizes the morphological disparity (also known as the Procrustes variance) among the three replicates of an individual for each scan type. The mean Procrustes variance for 3D scans was $1.34 \times 10^{-4}$ and $4.81 \times 10^{-5}$ for µCT scans. This is a significant difference ($p < 0.001$).

## Procrustes variance and repeatability

To compare the digitization error in each scanning device dataset, we calculated the Procrustes variance among the replicate triads of each individual. We found that Procrustes variance is significantly ($p < 0.001$) higher in 3D scans (mean = $1.31 \times 10^{-4}$) than in µCT scans (mean = $4.76 \times 10^{-5}$) (Fig. 6). This means that digitizations are more variable in 3D scans than in µCT which is consistent with decreased clustering in 3D scans relative to µCT scans in the PCAs (Fig. 4).

The repeatability for each scan dataset mirrored the Procrustes variance results. We found that the µCT scan dataset had a repeatability of 0.896 and the 3D scan data had a repeatability of 0.750 (Table 3A, Table 3D). This means operators are more successful at repeating their digitizations (i.e., landmark placements) with µCT scans than with 3D scans.

To test how different types of landmarks impacted repeatability, we calculated repeatability for combinations of landmark types for 3D and µCT datasets consisting of only the symmetric component of shape (Table 3). Because sliding landmarks depend on the placement of fixed landmarks (and patch points depend on both fixed and semi-landlandmark curves), we could not isolate each type of landmark's repeatability. The

**Table 3 Comparison of operator error in 3D scan and μCT scan datasets using Procrustes ANOVAs and repeatability scores.** The repeatability score (R) is a value that reflects the ease of digitizing in a repeated measure study design. It is calculated from the Procrustes ANOVA using formulas for the intraclass correlation coefficient. The Procrustes ANOVAs were found by subsetting the dataset by scan device and by landmark types and then performing separate generalized Procrustes and bilateral symmetry alignments. (A–C) Results from the 3D-only dataset. (D–F) Results from the μCT-only dataset. (A) and (D) show the repeatabilites from the entire landmark datasets of each scan device. (B) and (E) remove patch points. (C) and (F) contain only fixed landmarks.

**(A) 3D scan all landmarks including patches (n = 289)**

|  | Df | SS | MS | Rsq | F | Z | Pr(>F) | R |
|---|---|---|---|---|---|---|---|---|
| Ind | 18 | 3.53E−02 | 1.96E−03 | 0.826 | 10.0 | 16.0 | 0.001 | 0.750 |
| Residuals | 38 | 7.46E−03 | 1.96E−04 | 0.174 | | | | |
| Total | 56 | 4.28E−02 | | | | | | |

**(B) 3D scan fixed landmarks and semilandmarks (n = 203)**

|  | Df | SS | MS | Rsq | F | Z | Pr(>F) | R |
|---|---|---|---|---|---|---|---|---|
| Ind | 18 | 4.37E−02 | 2.43E−03 | 0.807 | 8.826 | 16.7 | 0.001 | 0.723 |
| Residuals | 38 | 1.04E−02 | 2.75E−04 | 0.193 | | | | |
| Total | 56 | 5.41E−02 | | | | | | |

**(C) 3D scan fixed landmarks only (n = 58)**

|  | Df | SS | MS | Rsq | F | Z | Pr(>F) | R |
|---|---|---|---|---|---|---|---|---|
| Ind | 18 | 6.90E−02 | 3.83E−03 | 0.749 | 6.30 | 16.6 | 0.001 | 0.639 |
| Residuals | 38 | 2.31E−02 | 6.09E−04 | 0.251 | | | | |
| Total | 56 | 9.21E−02 | | | | | | |

**(D) CT scan all landmarks including patches (n = 289)**

|  | Df | SS | MS | Rsq | F | Z | Pr(>F) | R |
|---|---|---|---|---|---|---|---|---|
| Ind | 18 | 3.46E−02 | 1.92E−03 | 0.927 | 26.9 | 18.4 | 0.001 | 0.896 |
| Residuals | 38 | 2.72E−03 | 7.15E−05 | 0.073 | | | | |
| Total | 56 | 3.73E−02 | | | | | | |

**(E) CT scan fixed landmarks and semilandmarks (n = 203)**

|  | Df | SS | MS | Rsq | F | Z | Pr(>F) | R |
|---|---|---|---|---|---|---|---|---|
| Ind | 18 | 4.33E−02 | 2.41E−03 | 0.921 | 24.7 | 19.0 | 0.001 | 0.888 |
| Residuals | 38 | 3.71E−03 | 9.76E−05 | 0.079 | | | | |
| Total | 56 | 4.70E−02 | | | | | | |

**(F) CT scan fixed landmarks only (n = 58)**

|  | Df | SS | MS | Rsq | F | Z | Pr(>F) | R |
|---|---|---|---|---|---|---|---|---|
| Ind | 18 | 6.28E−02 | 3.49E−03 | 0.893 | 17.6 | 20.2 | 0.001 | 0.847 |
| Residuals | 38 | 7.54E−03 | 1.98E−04 | 0.107 | | | | |
| Total | 56 | 7.03E−02 | | | | | | |

**Table 4  Symmetric Procrustes ANOVA with device and sex as factors to assess relative contribution of intra-specific variation to overall shape variation.** This Procrustes ANOVA allows comparison of the relative contribution to total variation from scan device and sex (R-squared column).

| | *Df* | SS | MS | Rsq | *F* | *Z* | Pr(>F) |
|---|---|---|---|---|---|---|---|
| Device | 1 | 2.99E−03 | 2.99E−03 | 0.0646 | 4.84 | 4.06 | 0.001 |
| Sex | 1 | 4.40E−03 | 4.40E−03 | 0.0952 | 7.14 | 4.96 | 0.001 |
| Residuals | 63 | 3.88E−02 | 6.16E−04 | | | | |
| Total | 65 | 4.62E−02 | | | | | |

analyses restricted to completely manually placed fixed landmarks always had the lowest repeatability of the three types of landmarks (Table 3C, Table 3F). Repeatability was always highest for the datasets including all three types of landmarks including the semi-automated semi-landmarks and the completely automated patch points (Table 3A, Table 3D). Higher repeatability in datasets with the sliding landmarks may result because the sliding smooths out user placement error across replicates.

## Analyses with a biological example: sexual dimorphism

A small subset of our dataset had sex information ($n = 11$; $f = 7$, $m = 4$), allowing us to perform a test of whether using different scan devices classify males and females according to shape with the same level of accuracy. Our Procrustes ANOVA on the symmetric component of shape variation using sex and device as factors found that shape differences due to device (Rsq = 0.0646) and sex (Rsq = 0.0952) are both significant ($p < 0.001$). Both factors have relatively small effect sizes, however, sex captures slightly more shape variation than device (Table 4). However, the between-group PCAs do not suggest marked sexual dimorphism to begin with (Fig. 7). Therefore, the subtlety of this biological signal could be the main reason for the small contribution of sex to total variation. Finally, we performed a cross-validation test on the between-group PCAs to assess which scan dataset can more reliably classify sexes based on shape (Table 5). The results show that in this case, 3D scans and µCT scans perform identically (overall classification accuracy = 63.6%).

## DISCUSSION

In this study, we contrasted very high resolution µCT scans with their extreme opposite: 3D surface scans of very small specimens. Our low versus high resolution datasets allowed us to assess whether the low resolution scans still allow defensible investigations of biological shape variation. We found that despite the low quality of the 3D scans, sufficient amounts of biological variation are present to perform, at the very least, typical interspecific comparisons. In datasets with only very slight intra-specific differences, more difficulties in distinguishing biological signal from the noise introduced by error during data collection. For example, the subtle sexual dimorphism in our small sample was only just distinguished. However, we present three considerations to make before using low resolution datasets. First, we found that variation due to scan device and digitizations is substantial relative to asymmetric variation. This makes low resolution datasets a poor choice for studies on asymmetry. Second, using 3D scans creates more random error due to increased

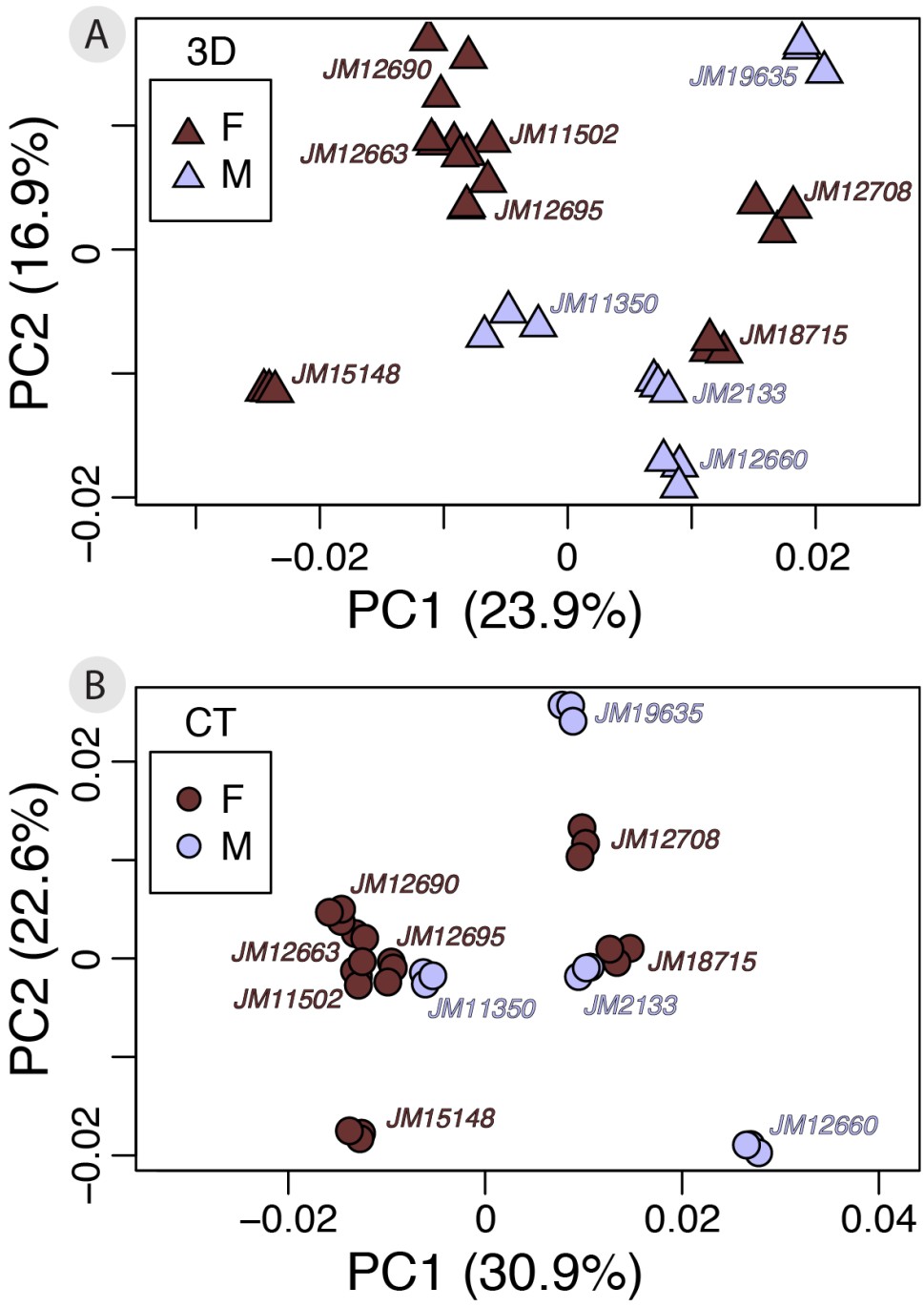

**Figure 7** ntra-specific variation as shown by PCAs of 3D (A) and μCT (B) scan datasets colored by sex. Each PCA provides an exploratory visualization of shape variation between males and females in our subsample with sex information ($n = 11$). Males ($n = 4$) are plotted in light blue and females ($n = 7$) are plotted in dark red. Results from the cross-validation test can be found in Table 5.

**Table 5  Between group PCA classification test to assess whether one scan device dataset performs better at identifying sexes based on shape.** This analysis averages shape among replicates, computes a between-group PCA separately for 3D and μCT datasets, and runs a cross-validation classification test. The results indicate whether one type of scan dataset is more successful at classifying males versus females based on the shape variation present in the dataset. It also returns a kappa statistic; a kappa value over 0.20 indicates "fair" agreement between the two datasets. Shape variation visualized by sex can be seen in Fig. 7.

| Cross-validated classification results in frequencies | | |
|---|---|---|
| 3D | f | m |
| f ($n = 7$) | 5 | 2 |
| m ($n = 4$) | 2 | 2 |
| CT | f | m |
| f ($n = 7$) | 5 | 2 |
| m ($n = 4$) | 2 | 2 |
| Cross-validated classification results in % | | |
| 3D | f | m |
| f | 71.4 | 28.6 |
| m | 50.0 | 50.0 |
| CT | f | m |
| f | 71.4 | 28.6 |
| m | 50.0 | 50.0 |
| Overall classification accuracy (%) | | |
| 3D | 63.6 | |
| CT | 63.6 | |
| Kappa statistic | | |
| 3D | 0.214 | |
| CT | 0.214 | |

landmarking difficulty, therefore care should be taken in landmark choice, and possibly landmarking software and operator choice. Digitization error may also be reduced by taking averages of repeated measurements (*Arnqvist & Martensson, 1998*; *Fruciano, 2016*). Third, using 3D scans also introduces a source of systematic error relative to μCT scans, therefore we recommend not combining them whenever possible (see also *Fruciano et al., 2017*), and especially in studies on small intra-specific variation. In summary, with a few precautions listed above, we expect that for studies with similarly sized skulls or similarly low resolution scans, the variation due to error will be sufficiently low for successful detection of interspecific shape differences.

## Measurement error and 3D scan reliability

Systematic error between the two scan devices is shown by consistent displacement patterns in the PCA. Indeed, across all three PC axes, the scans differ in how they measure concavity around the braincase, frontal, and ventral surface. This systematic pattern could suggest that the 3D scanner technology errs by adding volume to the digital specimen relative to the μCT scan but it could also be the other way around with the μCT scan distorting the images to reduce volume. Furthermore, even when using the symmetric component of shape, the percent of variation contributed by scan device is quite substantial at about

14.5%. Because scan device contributes this much to variation and because systematic error between scan device exists, researchers expecting very small variation due to biological sources would be advised not to combine 3D scan and μCT scan datasets.

While the two scan devices are usually comparable, using the low resolution 3D scans introduces more digitization error than the higher resolution μCT scans, which likely reflects increased user error due to lower resolution in 3D scans. This increased random error is reflected in both the larger point clouds of 3D replicates relative to μCT replicates in the PCAs, the higher Procrustes variance, and the lower repeatability score of 3D scans, particularly of manually-placed fixed landmarks. As expected, we found that the low resolution 3D scans were more difficult to landmark because key cranial features such as sutures and smaller processes were less distinct (Fig. 1). Nevertheless, our overall 3D scan repeatability score of 0.75 with symmetric data appears consistent with the literature: it is much lower than 3D scanned human-sized skulls—above 0.95 (*Badawi-Fayad & Cabanis, 2007*; *Fourie et al., 2011*) but it is approaching the range of 3D scanned macropodoids (e.g., kangaroos)—0.78–0.98, depending on device and landmark choice (*Fruciano et al., 2017*). This trend of decreasing repeatability with decreasing body size may reflect measurement error becoming a larger percentage of overall size (*Robinson & Terhune, 2017*). Relatedly, recent work has shown that excluding a few unreliable landmarks, or those with greater variability in placement, can significantly increase repeatability (*Fruciano et al., 2017*). This may be especially true for small specimens, for which small variations from the landmark location represent a larger percentage of their overall size.

Our repeatability tests on different combinations of landmark types suggest that fixed landmarks suffer the most from decreased resolution and the associated increased user error while patch points suffer the least. We interpret these results to mean that the (semi-) automatic placement of semi-landmark curves and patches is more consistent in placing points compared to a human operator placing fixed landmarks, regardless of whether the automatic placement is "correct" or not. It is important to note that while semi-landmarks were "semi-automated", the user still manually defined the curve they slid along for each specimen. Furthermore, this curve is bounded by user-placed fixed landmarks. Therefore, the increased repeatability with increasing automation could also be due to the increased degrees of freedom afforded to landmarks during sliding: fixed with zero degrees, semi-landmarks with one degree, and patch points with two. The sliding, by removing variation tangential to a certain direction, will reduce the variance in those points which will appear to vary less so it would be expected that these points will contribute less overall variation when combined with the fixed landmarks.

This study did not look at multiple operator error which can be considerable, particularly if difficult landmarks are included (*Fruciano et al., 2017*). If inter-operator error were combined with the resolution-driven measurement error found here, it is possible that biological signal would diminish to a degree that could not support even interspecific comparisons.

## Measurement error introduced by scanning device compared to biological variation

The challenge of any quantitative measurement study is to minimize measurement error introduced from various sources (in our case, device, resolution, and observer) relative to the "true" signal of biological variation. For subtle sources of biological variation, such as asymmetry, our results show that the error associated with collecting data from 3D scans contributes the same amount of variation as asymmetry. Therefore, a low resolution study of asymmetry with 3D scans would likely be unreliable unless appropriate arrangements were made to reduce error (*Fruciano, 2016*), whereas μCT scans may be more suitable for these types of studies. In the case of inter-observer error, which is another common source of measurement error, several studies suggest that interspecific variation can overwhelm inter-observer error such that this does not pose an issue with the correct interpretation of results (*Robinson & Terhune, 2017*).

In our test on the ability of different scan devices to classify according to sexual dimorphism, we showed that while variation contributed by each source was similar (and that from scan device slightly lower), both scan datasets presented a small sexually dimorphic pattern and supported the same classification performance. This suggests that 3D scans may even be acceptable for detecting some intra-specific patterns. However, this was a small sample ($n = 11$) and further studies with larger datasets would improve confidence for using 3D scans for intra-specific studies. Studies based on larger datasets might also be able to better highlight differences in classification performance between devices, if any. Nevertheless, it is promising that 3D scans and μCT scans performed equally even at such a small sample size for such a subtle intra-specific signal.

## Choosing a digitization device: 3D surface scanning versus μCT versus photogrammetry

With many options for digitizing 3D specimens available, decisions on the acquisition mode must consider price, scanning time, processing time, portability, and scan resolution. The one-off investment of a relatively high resolution 3D surface scanner such as the HDI109 provided a model portable enough to take on airplanes and with fast scanning and processing times. Our model took 10 min from starting the scan to the finished surface file, but note that larger specimens requiring multiple sub-scans will take longer. These fast acquisition times are an asset in collection efforts that rely on expensive and time-limited museum travel. For example, one of us (AEM) digitized over 100 individuals in one week using the same scanning protocol. However, the quality and speed of scanning varies by model; for example, other 3D surface scanners could take over 45 min to capture one specimen and may also require more effort to process scans (*Katz & Friess, 2014*).

Compared to 3D surface scanners, μCT scanners provide much higher resolution, which in this study translated into less measurement error. However, uCT facilities are not widely accessible, not mobile, and tend to be more expensive. Depending on the facility, μCT scanning involves transport to the facility, scanning either by the operator, processing scans into image stacks, and finally loading scans into specialized (and frequently high-cost) software to do the 3D reconstruction. These reconstructions can be time consuming

especially if the cranium needs to be separated from the mandibles. Finally, specimens need to be loaned from their collections for µCT acquisition, which requires specimen transport and curator permission and is particularly difficult when large numbers of specimens from distant locations need to be scanned.

This study did not investigate photogrammetry, which is another and increasingly popular method for digitizing 3D shape. This method uses software to align 2D photographs taken from many different views into a 3D file. Photogrammetry is much cheaper and more portable than 3D surface scanning since it only requires a camera of suitable resolution and very affordable photo-alignment software like Agisoft PhotoScan (Agisoft LLC, St. Petersburg, Russia; http://www.agisoft.com). The trade-offs are that in our experience, photogrammetry takes at least three times longer to acquire the photos, it involves higher risk of human error or inconsistency during photography, and it requires an order of magnitude more time to align the photos into a 3D digital file. While photo-alignment can be done at convenience after photography, the greater time required to capture enough photos may be a deciding factor for researchers with time limitations in museum collections. As for scan resolution, photogrammetry may perform better than 3D surface scanners in some cases (*Fourie et al., 2011*) or at least provide an acceptable alternative (*Katz & Friess, 2014*; *Muñoz Muñoz, Quinto-Sánchez & González-José, 2016*).

Scan resolution is not the only consideration when choosing a scan device as its unique requirements for 3D model processing may increase image noise and therefore landmarking difficulty. Compared to µCT scanning, 3D scans tend to be both noisier and require more model processing before 3D model export. Specifically, artificial smoothing and hole-filling may change the topography of the 3D mesh. Therefore, the comparison we have presented here is not just a comparison of resolutions but also a comparison of 3D model generation. The methods we provide in Supplemental Information 1 represent the settings we found to decrease noise, however, the software also required some model smoothing and hole-filling before export. We recommend that researchers take these additional sources of image modification into account during their landmark choice and study design.

## CONCLUSIONS

Here, we have shown that a 3D surface scanner can provide an acceptable alternative to a µCT scanner for assessing biological signal of 3D shape even in small specimens that are at the limits of 3D scanner resolution. Our analyses specifically showed that first, error contributes to a higher percentage of variation in 3D scan datasets than in µCT scan datasets of the same small specimens. As a result, we conclude that 3D scans are usually not appropriate for studies on very small sources of variation like fluctuating asymmetry. Second, we show that 3D scan datasets have a lower repeatability of landmark placement, especially for fixed landmarks, as compared to µCT scans. Relatedly, our comparisons of repeatability on data with asymmetry to the same data without asymmetry—i.e., having bilateral symmetry—support analyzing the bilaterally symmetrical data of landmarks from low resolution scans. Finally, we use a preliminary study of sexual dimorphism to suggest that despite elevated error and shape variance, bilaterally symmetrical datasets from 3D
scans can support male versus female classification based on small biological differences as well as μCT datasets can. In summary, while 3D scans are a promising alternative, exploratory pilot studies of measurement error like this one are advisable when practically possible (see also *Fruciano, 2016*).

Furthermore, the best 3D capture method will vary based on the study's design, expected effect size for the biological variation of interest, and the researcher's limitations on time, money, and travel. In addition to image resolution requirements, it is wise to assess the time it will take to capture and process each specimen as well as portability needs. We recommend a preliminary test on multiple devices–including surface scanners–to determine how levels of error compare to biological signal and whether there is substantial systematic error. Doing so may provide a defensible alternative to an expensive and time consuming large-scale acquisition of μCT scans including for studies on very small specimens.

**Abbreviations**

| | |
|---|---|
| **LM** | Landmark |
| **μCT** | Micro-computed tomography |
| **PCA** | Principal component analysis |
| **PC** | Principal component |
| **3D** | Three-dimensional |

# ACKNOWLEDGEMENTS

We would like to thank Cruise Speck for assistance with Viewbox software and Dr. Heather Janetzki for hosting us in the mammal collections at the Queensland Museum.

## Funding

This work was supported by an Australian Research Council Discovery Grant (DP170103227) to Vera Weisbecker and Matthew J. Phillips and by an International Postgraduate Research Scholarship and UQ Centennial Scholarship (00025B) to Ariel E. Marcy. There was no additional external funding received for this study. The funders had no role in study design, data collection and analysis, decision to publish, or preparation of the manuscript.

## Grant Disclosures

The following grant information was disclosed by the authors:
Australian Research Council Discovery Grant: DP170103227.
International Postgraduate Research Scholarship and UQ Centennial Scholarship: 00025B.

## Competing Interests

The authors declare there are no competing interests.

## Author Contributions

- Ariel E. Marcy conceived and designed the experiments, performed the experiments, analyzed the data, prepared figures and/or tables, authored or reviewed drafts of the paper, approved the final draft.
- Carmelo Fruciano conceived and designed the experiments, analyzed the data, prepared figures and/or tables, authored or reviewed drafts of the paper, approved the final draft.
- Matthew J. Phillips analyzed the data, approved the final draft.
- Karine Mardon performed the experiments, contributed reagents/materials/analysis tools, approved the final draft.
- Vera Weisbecker conceived and designed the experiments, analyzed the data, contributed reagents/materials/analysis tools, authored or reviewed drafts of the paper, approved the final draft.

## Data Availability

Morphosource: https://www.morphosource.org/Detail/ProjectDetail/Show/project_id/458.

## Supplemental Information

Supplemental information for this article can be found online at http://dx.doi.org/10.7717/peerj.5032#supplemental-information.

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
