# Peer review of "Low resolution scans can provide a sufficiently accurate, cost- and time-effective alternative to high resolution scans for 3D shape analyses"

_PeerJ, doi:10.7717/peerj.5032_

## Round 0.1 · original submission · Minor Revisions

Both reviewers are largely positive about your manuscript, with just a few suggestions for revision. In particular, reviewer 2 has some important comments regarding symmetric vs asymmetric variation and the types of landmark used. Once you've had a chance to respond to all these comments, I look forward to seeing a revised version of the manuscript.

·

Basic reporting

The article meets all requirements in this category.

Experimental design

This paper examined 3D landmark data produced from models created using different scan modalities (3D surface scanning, μCT scanning). While similar studies have been previously completed, all used relatively large specimens. Marcy and colleagues pushed the lower size boundaries of these scanning technologies in using a small rodent (delicate mouse) with cranium around 20 mm in length. The methods are described in sufficient detail including supplementary files with the raw data and stat code.

Validity of the findings

The authors found that biological variation in the crania was largely captured by the two different scanning modalities, but that the surface scanner did seem to produce more error. They make several recommendations for mitigating error and for working with lower resolution data.

This is a very well-conducted study and makes a necessary contribution to the literature on this subject. With the advent of morphosource and other repositories for 3D scan data, researchers are increasingly creating combined datasets and understanding how these data can and cannot be combined into different analyses is essential.

Additional comments

I made a few very minor comments on the attached pdf. Supplementary file “peerj-25642-Bilateral_Landmarks” needs more information about what the columns of data mean – it is unclear. The remaining supplementary files are fine.

This is a well-written and well-designed study! I look forward to seeing it in print.

·

Basic reporting

The manuscript is well written, the English being clear and professional. While most of the text is unambiguous two specific sections should be improved to ensure that a non-specialised audience can clearly understand what the authors are trying to explain. The sections where the writing could be improved include lines 147-149 and 306-307. In the first one, I understand that each specimen was scanned three times with each device, which gives a total sample of 114 three-dimensional models (not 114 replicates); however the sentence is somewhat ambiguous and complicated. Please, rephrase it in a more direct way. The sentence written in lines 306 and 307 is imprecise. I agree with the authors that the fact that the asymmetric variation is not significantly greater than the measurement error (ME) does not imply that symmetric variation is not (because asymmetric variation is usually smaller than symmetric variation). And therefore the fact that asymmetric variation is not significantly greater than the ME does not invalidate the use of the technique for the comparison of the symmetric component of shape. However, removing asymmetric variation does not allow investigating shape variation more confidently as state the authors. First of all, asymmetric variation is not strictly removed, but separated and analysed separately from the symmetric one. And second, comparing the ME with the symmetric or with the asymmetric component answers two different questions. The current phrase let the reader understand that analysing symmetric variation alone is “better” that including asymmetric variation. In fact, the more informative option is to include both components of form. However, because of the design of the Procrustes ANOVA, when the asymmetric variation is included the test compares asymmetric variation, but not symmetric, with measurement error.
The Introduction is very useful to put the reader into context and the literature is relevant and well referenced. However, while the main goal is clear, the specific objectives to accomplish it are not indicated in the text. In this part of the text, the authors reference the Figure 2, where the specific analyses and questions are highlighted. Indicating the specific objectives in the text will aid the reader to fully understand the aims of the authors.
Structure conforms to PeerJ standards. Figures are relevant, high quality and well labelled and described. Tables are also relevant but he number of decimals and the type of notation (scientific or decimal) should be standardized within and among tables.
Raw data and scripts are supplied which allow to replicate the study.

Experimental design

The study represents an original primary research within the scope of the journal. The research question is well defined and relevant. In this sense I found Figure 2 very informative and useful (but see comment in the above section regarding specific objectives). The research is an interesting contribution to the field of 3D modelling applied to geometric morphometrics. It adds information about the use of non-expensive methodologies, in particular low resolution 3D scanning, to obtain 3D models for the analysis of biological form. The investigation is rigorous and performed to high technical standards (but see General comments 1 and 2). The experimental design is appropriate to answer the questions formulated by the authors. Methods are described with enough detail and information (including raw data and R scripts), which would allow replicating the study in the future.

Validity of the findings

Data is robust and analyses are, in general, statistically sound (but see General comments 1, 3, and 4). Conclusions are very general and not linked in a one-to-one manner to research questions and supporting results. More precise Conclusions directly linked to research questions should be provided.

Additional comments

1. I have several doubts and comments about the last section of the Results (Analyses with a biological example: sexual dimorphism). In Table 4 variation due to sex is obtained together with residual variation. Sex should be analysed as a separate factor in the Procrustes ANOVA in order to test if differences between sexes exist. If differences between sexes are not significant, the cross-validation analysis is not justified and the whole section loses its usefulness, and should be removed from the manuscript. In order to compare the two methodologies in detecting a source of variation, it seems reasonable to assess the effect of a significant biological factor. In this sense, the PCA and the cross-validation analyses suggest that sex differences are at most very subtle. Moreover, the sample size is not the most appropriate to detect subtle differences between groups.
2. Another important point is the use of the term “interspecific” in the title. The title suggests that the utility of low resolution scans for interspecific comparisons have been tested, and this has not been done. I would change the title.
3. In lines 221-224 the authors state that “digitization error is large compared to components of asymmetric variation, even a single device…”. This sentence, which is based in the general ANOVA, is somewhat confusing because seems to suggest that both devices are equally affected by error. However, how digitization error affects the analysis of asymmetric variation in each method has not been assessed. In order to assess how digitization error affects each methodology, ANOVAs including the factor side should be performed separately for Micro-CT and 3D scans.
4. In lines 278-280 the authors say that “This means operators have an easier time repeating their digitisations (i.e. landmark placements) with Micro-CT than with 3D”. However, three types of landmarks have been used in the study: “true” landmarks, semi-landmarks, and patch points. It seems that the operator have played a different role in placing each type of landmark. While “true” landmarks have been placed manually, semi-landmarks have been recorded semi-automate, and patch points fully automate. How this differences affect ME in each methodology have not been assessed. It would be interesting to assess if some type of landmark is more affected by device error or observer error than other types. Performing separate ANOVAs for each type of landmark and method will answer these questions.
5. In lines 251-252 the authors say that “3D scans score higher than their Micro-CT scan replicates on both PC1 and PC2”. Please, confirm this. I see the •D scan points more left (negative) and below (negative) than Micro-CT points.
6. In line 260 the authors say “On the PC3 axis, Micro-CT scan replicates consistently score higher”. Again, please, confirm this. I see a lot of the Micro-CT points below 3D scan points.
7. Move lines 249-251 to Discussion.
8. Move lines 262-267 to Discussion.
9. Lines 120-121: change (Pseudomys delicatulus, J Gould, 1842) by Pseudomys delicatulus (Gould, 1842).
10. Line 204: change (Fig. 2c) by (Fig. 2d).
11. Line 215: change (fluctuating and directional) by (directional and fluctuating).
12. Line 251: which Table is Table TK?

---

## Round 0.2 · Minor Revisions

I am happy that you have addressed all the reviewer comments and the manuscript is looking great now. There are just a few minor issues to correct, but these are all typographical and stylistic - there are no substantive changes to make to the content. Reviewer 2 has listed a few comments in their review, and I have made some suggested revisions in the attached PDFs. I don't think any of these will take long to correct, so I look forward to seeing a revised copy of the manuscript in the very near future.

·

Basic reporting

The authors have rephrased properly the parts of the manuscript that were somewhat ambiguous and have clearly defined the objectives of the work.

Experimental design

No additional comments.

Validity of the findings

In the new version of the manuscript the conclusions are clearer and more directly linked to research questions than in the previous version.

Additional comments

The authors have dealt satisfactorily with all the comments that I made in the previous version of the manuscript, adding additional analyses when necessary. Consequently, I consider that the manuscript is suitable for publication in PeerJ after some minor changes. Please, see comments below.

1. The reference list should be revised carefully. Several errors have been detected, of which some are
listed below (the list is not exhaustive):
• The reference of Muñoz-Muñoz et al. (2016) has been entered twice. Remove the second one
(lines 577-579).
• Change “Munoz-Munoz” by “Muñoz-Muñoz” in lines 81, 107 and 574
• Change “Perpinan” by “Perpiñán” in lines 107 and 574
• The reference of Yezerinac et al. (1992) is all written in capital letters (lines 625-627). Please
correct it.
• Line 608, italicise “Ursus spelaeus”.
2. Lines 248-250 would be better placed in the Discussion.

---

## Round 0.3 · accepted · Accept

Thanks for making those last few changes. I'm now pleased to accept this manuscript for publication. Looking forward to seeing it published!

#